# Axl contributes to efficient migration and invasion of melanoma cells

**Hanshuang Shao**[1,2], **Diana Teramae**[1,2], **Alan Wells**[ORCID][1,2]*

**1** Department of Pathology, University of Pittsburgh, Pittsburgh, Pennsylvania, United States of America,
**2** Pittsburgh VA Health System, Pittsburgh, Pennsylvania, United States of America

* wellsa@upmc.edu

**Data Availability Statement:** All relevant data are within the paper and the original blots are supplied in the Supporting information file as per PLoS ONE guidance.

**Funding:** This work was supported by a Merit Award from the Veterans Administration

## Abstract

Axl, a member of the TAM receptor family has been broadly suggested to play a key role in tumor metastasis. However, the function of Axl in the invasion and metastasis of melanoma, the most lethal skin cancer, remains largely unknown. In the present study, we found that melanoma cell lines present variable protein levels of Axl and Tyro3; interestingly, MerTK is not noted at detectable levels in any of tested MGP (metastatic growth phase) cell lines. Treatment with recombinant human Gas6 significantly activates Akt in the Axl-expressing WM852 and IgR3 lines but just slightly in WM1158. IgR3, WM852 and WM1158 demonstrate different autocrine signaling. Knockdown of Axl by siRNA or the treatment with Axl-specific inhibitor R428 dramatically inhibits the migration and invasion of both IgR3 and WM852 *in vitro*. These findings suggest that Axl enhances the invasion of melanoma cells.

## Introduction

Melanoma originating from the pigmented melanocytes in the basal layer of the epidermis is the most lethal skin cancer although it accounts for only 1% of all patients with skin malignancies [1, 2]. Primary melanoma can be successfully surgically excised if the tumor cells grow radially. However, the 5-year relative survival rates of patients with melanoma dramatically decrease once the disseminated tumor cells invade into and especially through the dermis. Thus, vertical invasiveness is a harbinger of poor outcomes for these patients.

The overall prognosis of patients has dramatically improved in the past decades due to the availability of early screens, advanced surgeries, targeted therapy, and immune checkpoint inhibitors [3, 4]. For example, specific inhibition of BRAF leads to a strong response and survival benefit in most cases. However, only a small fraction of patients with advanced melanoma can achieve long-term survival [5, 6]. The major reason is likely due to the acquisition of drug resistance and in the survival signaling from the metastatic tumor microenvironment. For example, the activation status of PTEN controls the AXL/AKT axis mediated-resistance to BRAF inhibitor (BRAFi) [7]. Therefore, determining the functions of the key proteins which lead to the initial invasion, of melanoma is still urgent and necessary.

TAM receptor family of tyrosine kinases consists of Axl, Tyro3 and MerTK. Among three members, Axl is the most studied and has been shown to be elevated in many cancers

(BX003368 to AW). The funders had no role in study design, data collection and analysis, decision to publish, or preparation of the manuscript.

**Competing interests:** The authors have declared that no competing interests exist.

including melanoma, breast, lung, ovary, pancreas, and prostate [8–12]. Axl can be activated when its ligand, growth arrest-specific protein (Gas6), binds on resulting in a cascade of downstream signaling pathways [13]. Gas6 consists of three major domains [14]. Gas6/Axl mediates migration and invasion in many cancer types such as breast cancer, prostate cancer, and osteosarcoma [15–17]. We previously revealed that Tyro3 is important in melanoma migration via its phosphorylation and role in the regulation of ACTN4 [18]. In this study, we queried whether Axl could contribute to melanoma invasiveness.

## Materials and methods

### Reagents and cell culture

Validated Axl (Cat #: 4390824) and scramble siRNA (Cat #: 4390843), Transfection reagents Lipofectamine RAN/MAX 2000 (Cat #: 13778–150) and Lipofectamine 2000 (Cat #: 11668–019) were purchased from Life Technologies (Grand Island, NY). Axl (Cat #: TF320269 set A-D) and scramble (Cat #: TR30007) shRNAs were purchased from Origene (Rockville MD). Recombinant hGas6 (Cat #: 885-GSB) was purchased from R & D system (Minneapolis MN). Recombinant human epidermal growth factor (EGF) (Cat #: E9644-.2MG) was purchased from Sigma (St. Louis, MO). Axl inhibitor R428 (Cat #: A8329) was purchased from ApexBio (Houston TX). G418 (Cat #: G5005), was purchased from Teknova (Hollister CA). Antibodies including EGFR (Cat #: 4267S), pEGFR(Y1173) (Cat #: 4407S), pAkt(S473) (Cat #: 3787S), pERK (Cat #: 4370S), pP38 (Cat #:4511S), GAPDH (Cat #: 5174S), pAxl (Y702) (Cat #: 5724S) were purchased from Cell Signaling Technology. Monoclonal antibodies against Axl (Cat #:sc-166269), Tyro3 (Cat #: sc-166359) and ACTN4 (Cat #: sc393695) were purchased from Santa Cruz Biotechnology (Dallas, TA). MatriGel 24w transwells (Cat #: 354480) were purchased from Corning (Corning NY). Melanoma cell line IgR3, WM852, WM1158, WM983A, WM983B, FEMX and A375 were cultured in a medium containing 3 parts of DMEM (1gL$^{-1}$ glucose) (Cat #: 10-014-CV) and 1 part of Leibovitz's 15 (L15) (Cat #: 11415–064, Gibco, Billings, MT) with 10% fetal bovine serum (Cat #: 100–016, GeminiBio, West Sacramento, CA) and 1x pen/strep antibiotics (Cat #: 15140–122, Gibco). Melanoma cell line 1025Lu were maintained in RPMI 1640 (Cat #: 61870–036, Gibco) medium with 10% fetal bovine serum and 1x pen/strep antibiotics. Melanocytes were grown in DermaLife M media (Cat #: LM-0004) in the presence of growth factors and chemical components (Cat #: LS-1030, Lifeline Cell Technology, Frederick, MD).

### Immunoblotting

Cells grown in 6-well tissue culture plates were transfected with siRNA or treated with Gas6 or EGF prior to lysing in RIPA buffer containing 1x protease inhibitors cocktails set V. The lysate was incubated on ice for 5 min prior to sonicating briefly. Samples were collected to microcentrifugation tubes and then centrifuged at 13,000g at 4˚C for 30 min. After transferring the supernatant to new tubes, the concentration of total soluble proteins of each sample was determined using BCA™ Protein Assay (Cat #: 23227, Thermo Scientific™ Pierce™, Rockland, IL). To run a SDS gel, ten micrograms of total proteins from each sample was mixed with one fifth volumes of 5x SDS sample buffer with β-mercaptoethanol to get a running sample containing 1x SDS sample buffer. Protein samples were denatured by boiling for 3 min and then applied to SDS gel with different concentrations of acrylamide based on the molecular size of interest protein. Then, separative proteins in SDS gel were transferred to polyvinylidene difluoride (PVDF) membrane. The membrane was incubated with 5% fat free milk at room temperature for half followed by incubation with indicated primary antibodies in cold room overnight. Next day, the membrane was washed 3 times with 1x TBS buffer containing 0.1% Tween 20

for 5 min each. Then the membrane was incubated with secondary antibody at room temperature for 45 min followed with throughout washing. Finally, interest proteins on PVDF membrane were developed using enhanced ECL reagents.

## Transfection and selection of stable cell lines

In a six-well plate, $4x10^5$ cells were seeded in each well and cultured in complete growth medium overnight to get about 60% cell density at transfection. Next day, the medium was replaced with fresh Opti-medium with low serum followed by an addition of a complex of Lipofectamine RNA/MAX and siRNA for 4h incubation. Then, cells were incubated at 37˚C in a humidified incubator with 5% $CO_2$ for 48h prior to further. For selecting stable expressing cells, transfected cells were further cultured in complete growth medium containing 1400 μg/ml G418 until monoclonal colonies were big enough to be picked up.

## *In vitro* scratch wound assay

Confluent cells grown in six-well plate were slowly scratched with a rubber scraper to create a wound area at about 3 mm in width. After completely washing away the floating scratched cells, two lines vertically with scratched edge at the bottom of each well were marked. Then, images involving each mark line were taken as time 0h. After 24h incubation, images at same positions were retaken. For the treatments of Gas6 and EGF, cells were starved with quiescence media containing 0.5% dialyzed fetal bovine serum overnight prior to scratching. Finally, the width of each wound along the mark lines area was measured using Photoshop software. The relative migratory rates were the wide difference between time 0h and 24h.

## Transwell invasion assay

Cells were detached from culture plate with trypsin treatment and collected into a 15 ml Falcon tube. After centrifugating at 250g for 5 min, cells were resuspended in growth medium at a density of 50,000 cells/ml. Then, half a milliliter of cells was applied to a well equilibrated MatriGel transwell for an additional 48h incubation. For Gas6 treatment, cells were starved in quiescence medium containing 0.5% dialyzed FBS overnight and appropriate amount of Gas6 was added to the quiescence medium in the bottom chamber of transwell. After aspirating the medium in transwell chamber, all cells retaining on the top surface of MatriGel were removed using a loosen cotton swab. Then, cells invaded through the MatriGel and membrane and attached on the other side of membrane were fixed followed by staining with 0.5% crystal violet or permeabilized using Triton X-100 and staining with DAPI. Finally, cells were imaged and counted at different area under microscope.

## Results

### TAM and EGFR family receptor profiles of melanoma cell lines

We, and others have previously reported that melanoma cell migration is promoted by signals through receptors belonging to the EGFR/HER and TAM families [18–20]; These receptors have been implicated in metastatic progression across many different tumor types by driving tumor cell motility that enables invasion and dissemination. Vertical migration through the dermal layer is the critical step in developing metastatic melanoma.

To understand the signals that impel primary melanoma tumors to become metastatic, we determined the expression of TAM family members (Axl, Tyro3 and MerTK) in melanoma cell lines including vertical growth phase (VGP) and metastatic growth phase (MGP) using immunoblotting. Compared to melanocytes, all eight melanoma cell lines present increased

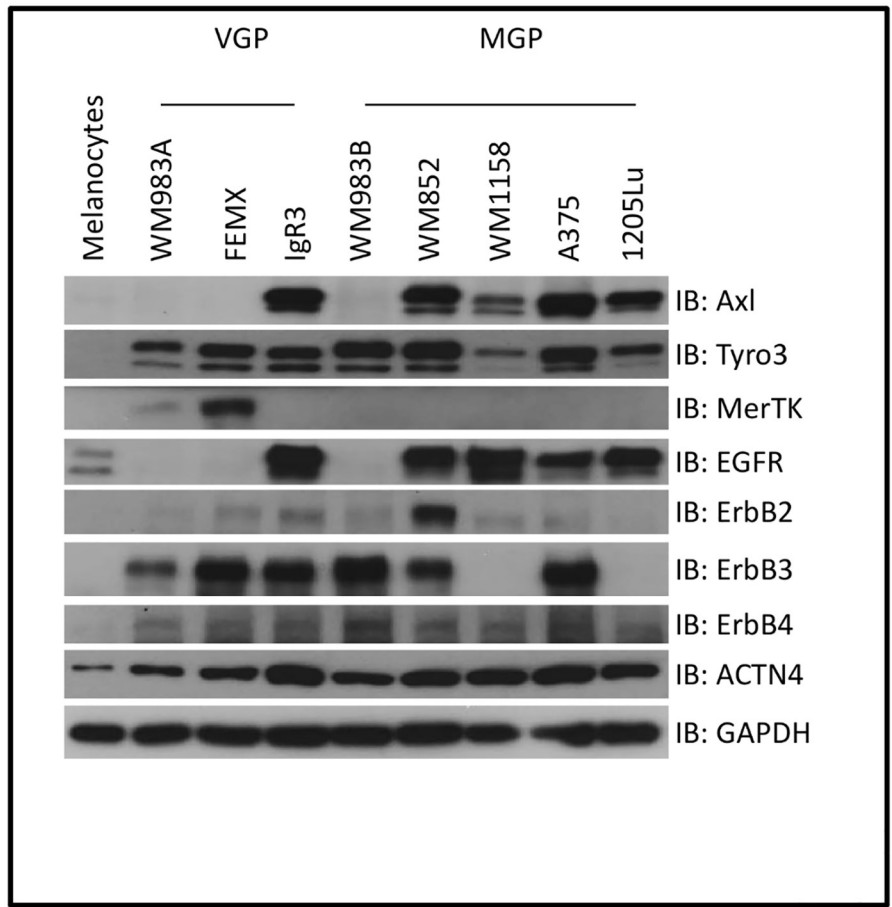

**Fig 1. TAM and EGFR family receptor profiles of melanoma cell lines.** VGP and MGP melanoma cells were lysed in RIPA buffer and soluble proteins were separated by SDS-PAGE for immunoblotting against indicated antibodies. Shown are representative blots of three experiments.

expression of Tyro3 (Fig 1). Interestingly, the Axl level in four out of five MGP cell lines (except for WM983B) is significantly elevated but only one cell line IgR3 from VGP shows more Axl expression. MerTK, a lesser studied TAM member, is not readily detectable in melanoma cell lines except in two VGP cell line 983A and FEMX. As there has been reported associations between Axl and EGFR family members and this has been proposed as a means to overcome resistance to anti-EGFR therapy [20–24], we queried the protein expression level of EGFR/HER family. As shown in Fig 1, EGFR is dramatically upregulated in four MGP cell lines (except for WM983B) and one VGP cell line IgR3 which are all Axl positive. ErbB2 is slightly increased in all tumor cell lines except for a significant increase in WM852 and undetectable in 1205Lu, similar to negligible if any levels in melanocytes. Surprisingly, all three VGP cell lines present a significant increase in ErbB3 expression while two MGP cell line WM1158 and 1205Lu do not express any ErbB3. The expression of ErbB4 is detectable in all eight cell lines but at low levels requiring extensive exposure times. Interestingly, while the expression pattern appears to be cell line specific, the more invasive and metastatic lines present upregulation of Axl and EGFR (except for WM983B), which appears absent in the VGP lines, with the exception of IgR3.

As we have reported that melanoma invasion involves the cytoskeleton to membrane linker protein alpha-actinin-4 [19] and that this is a downstream target of both receptor families [18, 25, 26], we catalogued its level. We found that this target is elevated in the eight cancer cell lines queried. The levels in the MGP lines were somewhat higher than in the VGP lines with the exception of IgR3 again expressing levels similar to MGP lines.

## Axl-dependent activation of intermediary signaling pathways by Gas6

Axl has been shown to play a key role in the acquisition of therapeutic drug resistance via activation of intermediary survival signals including the PI3kinase-Akt pathway [22]. These same pathways are involved in progression related cell behaviors including cell locomotion [27] and cell phenotype differentiation [28]. We queried whether the TAM-binding growth-arrest specific protein (Gas6) triggers differential activation of intermediary signaling pathways in different cell lines. We selected IgR3 and WM852 from VGP and MGP groups, respectively, as these lines express more Axl, Tyro3 and all four HER family members. We treated both IgR3 and WM852 cells, both starved in quiescence media containing 0.1% dialyzed FBS overnight, with recombinant Gas6 at a concentration of 200 ng/ml for 10 min. As shown in Fig 2A, we detected significant phosphorylation of Axl at Y702 in both IgR3 and WM852 which resulted in downstream activation of Akt. This is consistent with data found by [16, 17]. p38 is just slightly activated in WM852 and not at all in IgR3 when Gas6 was applied. Surprisingly, we did not observe an increase in pERK in either IgR3 or WM852 in response to Gas6. Of note, there may be some autocrine signaling as IgR3 presents higher basal level of pAkt than WM852 (NT lane). On the other hand, WM852 shows high basal p38.

As Axl has been shown to interact with EGFR in breast cancer cells and glioblastoma multiformes [23, 29], we were interested in whether Gas6 can activate EGFR in both IgR3 and WM852 cells in which both Axl and EGFR are highly upregulated. We did not observe phosphorylation of EGFR at Y1173 in WM852 cells and negligibly at best in IgR3 cells; the EGFR was functioning as shown by increased phosphorylation when exposed to 10 nM EGF for 10 min. Combination of Gas6 and EGF did not enhance pEGFR in WM852 and but did so significantly in IgR3.

To determine whether Gas6 activates Axl specifically and not other TAM family receptors, we downregulated Axl expression using siRNA (Fig 2A). Knockdown of Axl eliminated Gas6-mediated activation of Akt in both cell lines. To further confirm Gas6 specifically activates Axl, we applied Gas6 on another melanoma cell line WM983B in which Axl is undetectable. As shown in Fig 2B, pAkt levels are not affected by application of Gas6 to WM983B compared to IgR3 and WM852. This result also suggests Gas6 activation in IgR3 and WM852 is not through Tyro3 as WM983B expresses more Tyro3 than IgR3 and WM852.

## Gas6 enhances the migration and invasion of melanoma cells

The above data show that Gas6 leads to activation of intermediary signal pathways that contribute to cell migration. Next, we were interested in whether Gas6 promotes the migration of melanoma cells. As growth media contains stimulatory growth factors, we optimized the amount of dialyzed FBS and Gas6 that could demonstrate a contribution of signaling through Axl. As shown in Fig 3A, we observed a significant activation of Akt by Gas6 in all treatments. Although the lower concentration of FBS (0.1%) presented trace of pAkt in the absence Gas6, the contents of both pAxl and pAkt in the presence of Gas6 were also lower than those in other treatments. High concentration of FBS such as 2% led to the highest activation of Axl and Akt but the untreated (NT) cells showed higher basal activation of Axl and Akt. Therefore, we chose 0.5% FBS for optimizing the concentration of Gas6. We found that the phosphorylation

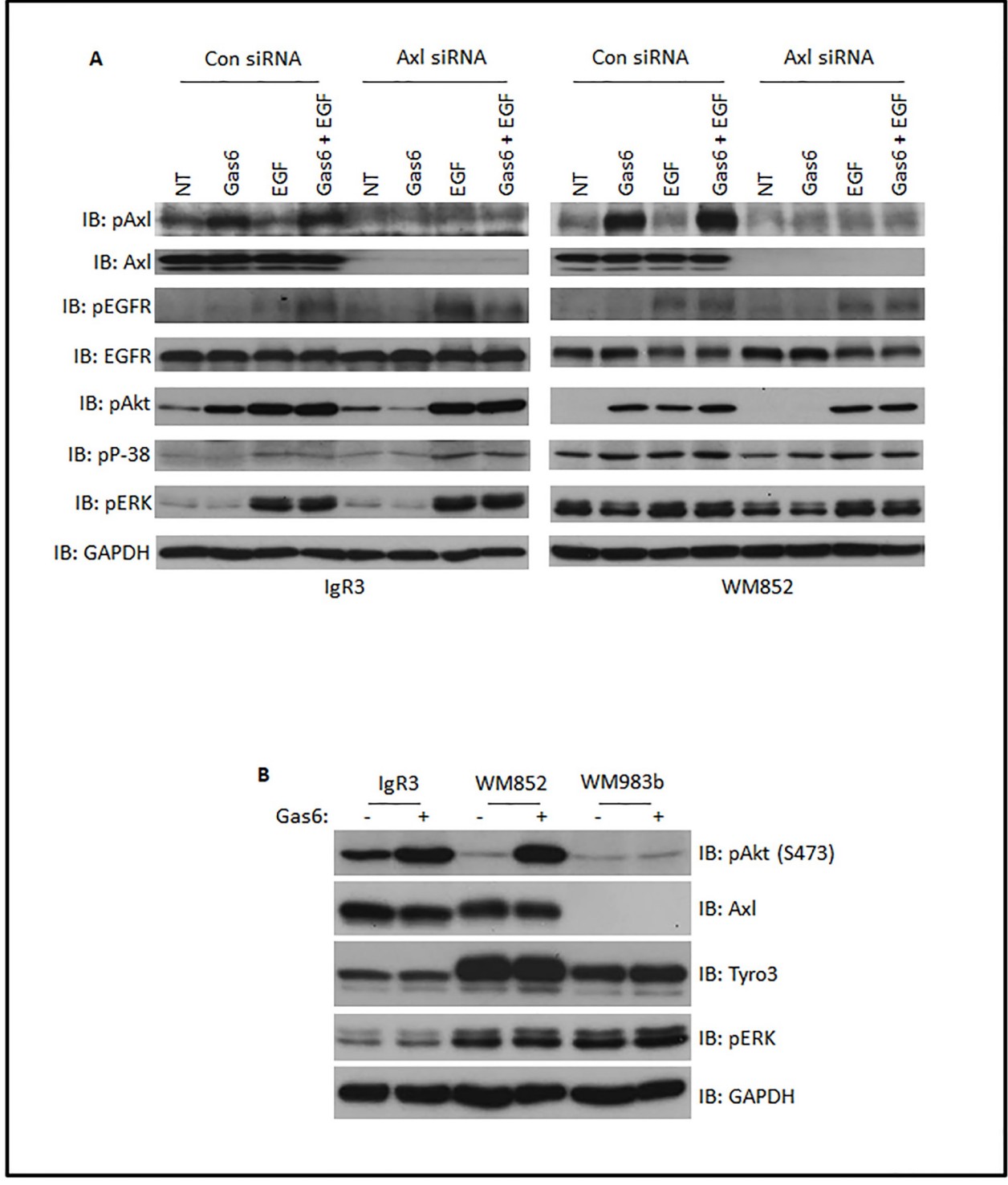

**Fig 2. Gas6-mediated activation of Axl in melanoma cells.** (A) Indicated melanoma cell lines were transiently transfected with control or Axl siRNA for 2 days prior to starving in quiescence media for 16h. Quiescent cells were then stimulated with indicated factors at the concentrations of 10 nM (EGF) and 200 ng/ml (Gas6) for 10 min followed by lysis in RIPA buffer for immunoblotting using the indicated antibodies. (B) Immunoblotting of indicated proteins extracted from quiescent melanoma cell lines treated with or without 200 ng/ml Gas6 for 10 min. NT: non treatment. Shown are representative blots of three experiments.

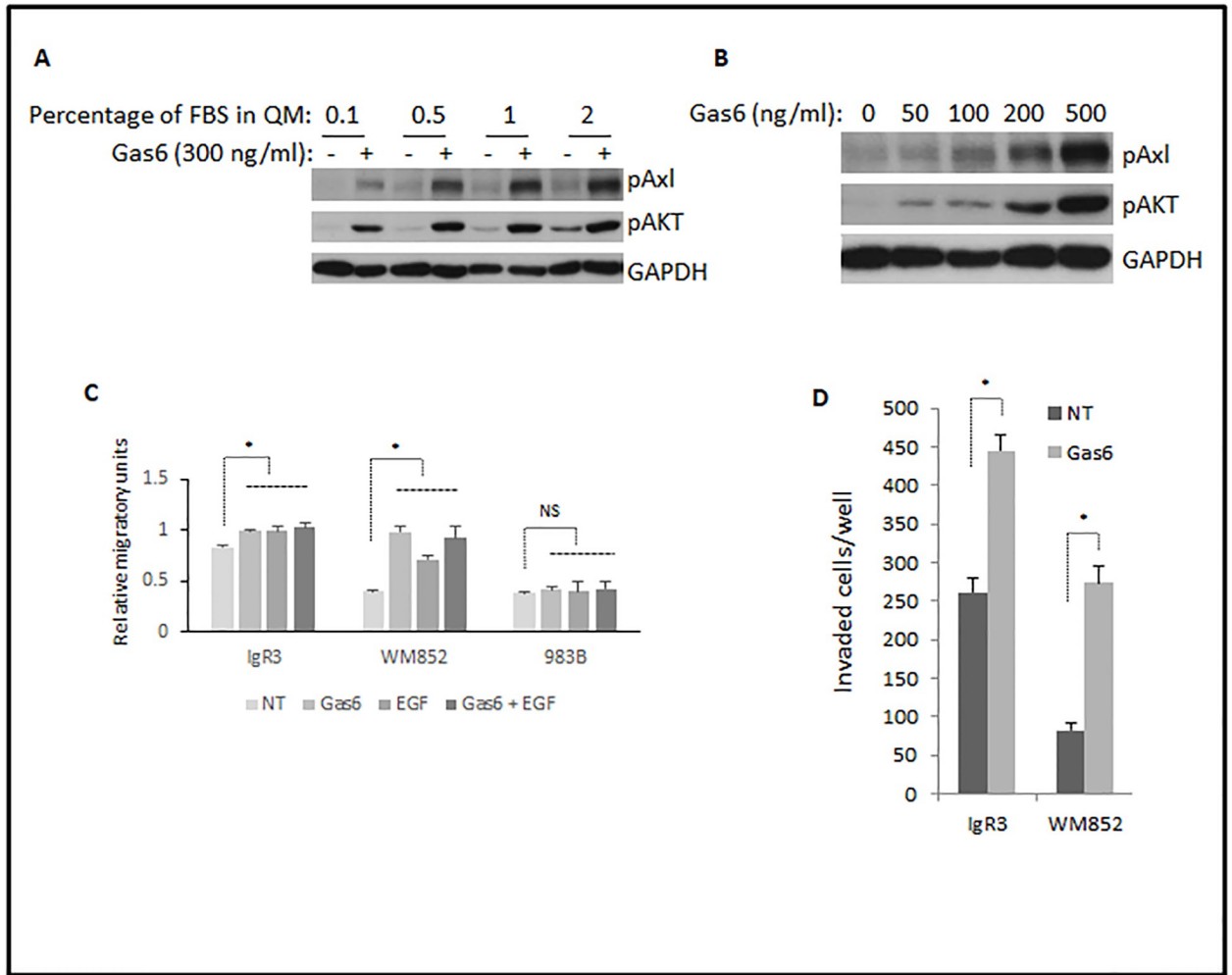

**Fig 3. Gas6 promotes migration and invasion of melanoma cells *in vitro*.** (A) Immunoblotting of indicated proteins extracted from WM852 cells quiesced in quiescent media containing indicated percentage of dialyzed FBS and stimulated with 300 ng/ml Gas6 for 15min. (B) WM852 cells were quiesced in 0.5% FBS quiescent media for 16h and then stimulated with indicated concentration of Gas-6 for 15min. Immunoblottings were performed using indicated antibodies. Shown are representative of three experiments. (C) Quantitative results of relative migratory speed of indicated melanoma cell lines stimulated with indicated factors for 24h. (C) Quantitative results of invaded IgR3 and WM852 with or without an attraction of 500 ng/ml Gas6 for 48h. Data are mean of ± SD of three independent experiments. Statistical analysis was performed using Student's t-test. * $P < 0.05$, NS: not significant.

levels of both Axl and Akt were gradually enhanced when increasing the concentration of Gas6 (Fig 3B).

We incubated melanoma cells in quiescence media containing 0.5% dialyzed FBS with 500 ng/ml of Gas6 for 24h to determine the effect of Gas6 on the migration of melanoma cells. As shown in Fig 3C, Gas6 dramatically promoted the migration of WM852, while moderately lifting the migration of IgR3 though IgR3 cells present higher level of migration in the absence of added Gas6. Migration induced by EGF was a positive control and showed that Gas6 was as active as EGF in driving motility. Interestingly, Gas6 and EGF were not additive. As a negative control, 983B cells showed no increase in motility by either Gas6 or EGF, as these cells do not express appreciable levels of either receptor Axl or EGFR (Fig 1). These findings suggest that Axl plays an important role in Gas6-mediated enhanced migration of IgR3 and WM852 (Fig 2A).

To further determine if Gas6 promotes invasion of IgR3 and WM852, we placed IgR3 or WM852 cells starved with 1.0% FBS quiescence media on the top of MatriGel and allowed the cells invade for 48h. We increased the concentration of dialyzed FBS in this assay due to extended time exposure and to optimize cell attachment and invasion through the MatriGel (S1 Fig). As we expected, 500 ng/ml Gas6 plus 1% FBS dramatically promoted the invasions of both IgR3 and WM852 (Fig 3). These data provide a role for Gas6 in promoting melanoma invasion through a matrix barrier consistent with prior findings suggesting such a role [16, 17, 30].

## Axl contributes to intrinsic migration and invasion

As Axl is abundant in IgR3 and WM852 cells and high density of receptors with intrinsic tyrosine kinase activity can lead to ligand-independent activation [31–34], we queried whether Axl contributes to the migration and invasion of these two cell lines even in the absence of Gas6. We transiently transfected these two cell lines with validated siRNA prior to performing monolayer wound scratch and MatriGel transwell invasion assays. The Axl siRNA dramatically downregulated the expression of Axl in both IgR3 and WM852 cells while the protein level of Tyro3 was not affected by Axl siRNA providing for the specificity of the siRNA (Fig 4A).

Axl siRNA significantly limited the migration of IgR3 and WM852 cells (Fig 4B, NT, black columns). To attribute this effect to Axl signaling, we observed that R428, an Axl-selective inhibitor at a concentration of 1 μM significantly reduced the migration IgR3 and WM852 cells transfected with control siRNA, but only slightly if at all cells in which the siRNA downregulated Axl. The reduced cell motility of both IgR3 and WM852 cells was not due to the possibility of R428 reducing cellular viability (S2 Fig). WM852 were almost completely killed in the presence of 10 μM of R428 for 24hr while IgR3 presented partial survival (S3A Fig). This difference in tolerating high concentration of R428 was probably due to the different level of activated Akt (pAkt at S473) even in the absence of added stimuli (Fig 2A and 2B, "NT" & "-"lanes). The basal level of pAkt in IgR3 was significantly higher than that in WM852. Axl siRNA did not affect the basal level of pAkt in IgR3 which explains why Axl siRNA did not increase the sensitivity of IgR3 cells to R428 (S3B Fig). In contrast, Axl siRNA made WM852 more sensitive to low concentration of R428 such as 2.5 μM and 5 μM (S3B Fig).

Next, we performed a MatriGel transwell invasion assay to determine whether Axl siRNA and R428 affect the invasion of IgR3 and WM852 cells. As shown in Fig 4C and 4D, both Axl siRNA and R428 dramatically reduced the number of invaded cells. The combination of Axl siRNA and R428 was not additive suggesting that R428 functioned specifically via Axl receptor. Taken together with prior reports on Axl functioning [35, 36], Axl is required for migration and invasion of melanoma IgR3 and WM852 cells *in vitro*.

To further confirm the role of Axl in migration and invasion in melanoma cells, we constructed a set of Axl shRNA vectors to generate stable cell lines of IgR3. After selection in puromycin, we obtained several stable mono-clones in which Axl was dramatically downregulated, but Tyro3 expression was not affected (Fig 5A). Monolayer scratch assays showed that all Axl stable clones migrated significantly slower compared to clones made with a scramble shRNA (Fig 5B). The invasion of all stable clones was dramatically less than cells transfected with the scramble shRNA clone (Fig 5C and 5D). These data further confirm that Axl is a key component for enhanced migration and invasion of melanoma cell line IgR3.

## Discussion

The lethal event in melanoma progression is invasion through the dermis, allowing for dissemination of the tumor cells to distant organs. The drivers of such would thus be targets to

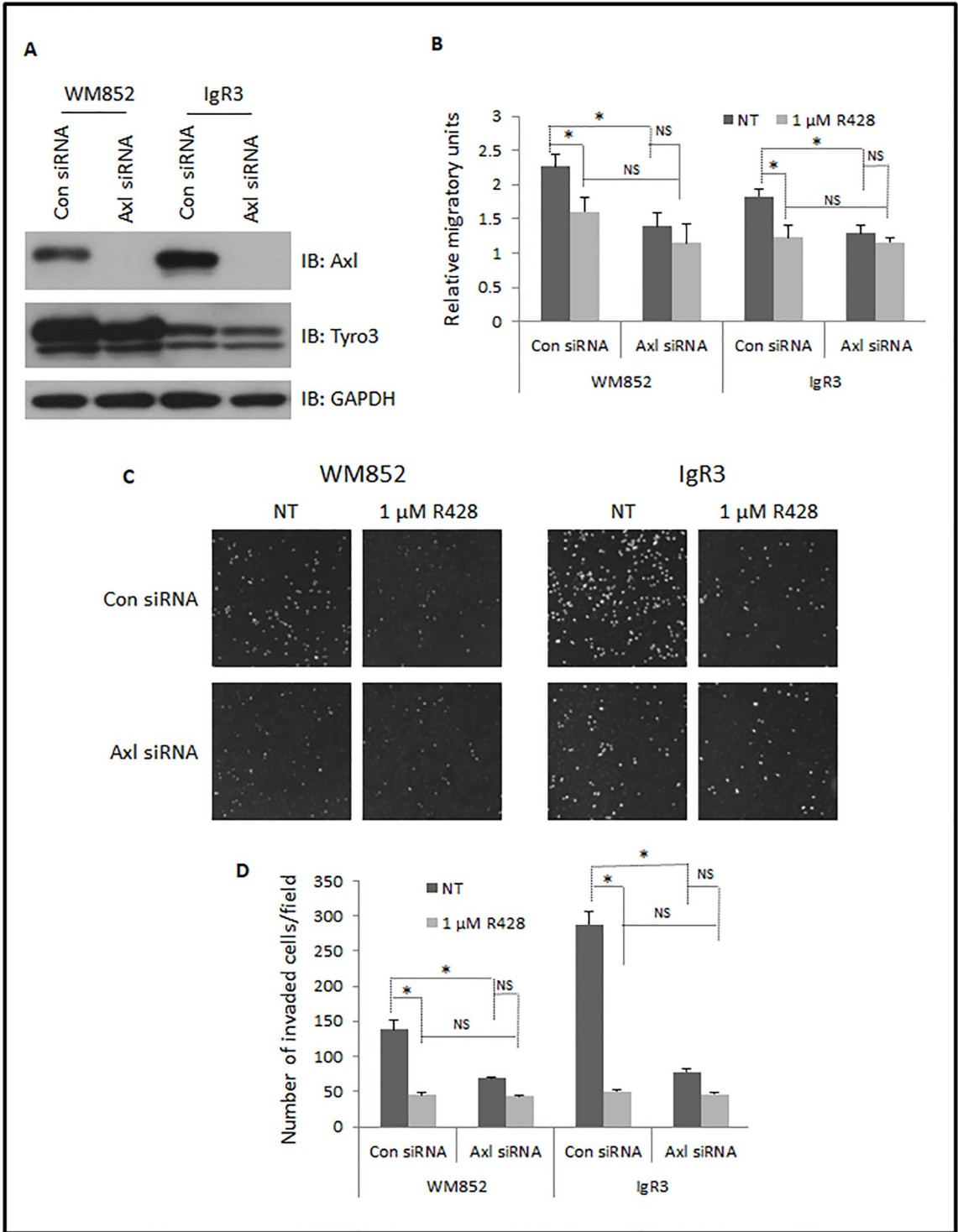

**Fig 4. Transient silencing of Axl using siRNA reduces migration and invasion of melanoma cells.** IgR3 and WM852 cells were transiently transfected with Axl siRNA for 48h. (A) Immunoblottings were performed using indicated antibodies to show selective downregulation of Axl. (B) Quantitative results of relative migratory speed of transfected cells with or without 1 μM of R428 treatment. (C) Representative images of DAPI-stained cells invaded through MatriGel and retained on the bottom of transwell membrane. (D) Quantitative results of invaded cells. Shown are representative blots and pictographs of three experiments. Data are mean of ± SD of three independent experiments. Statistical analysis was performed using Student's t-test. * P < 0.05, NS: not significant.

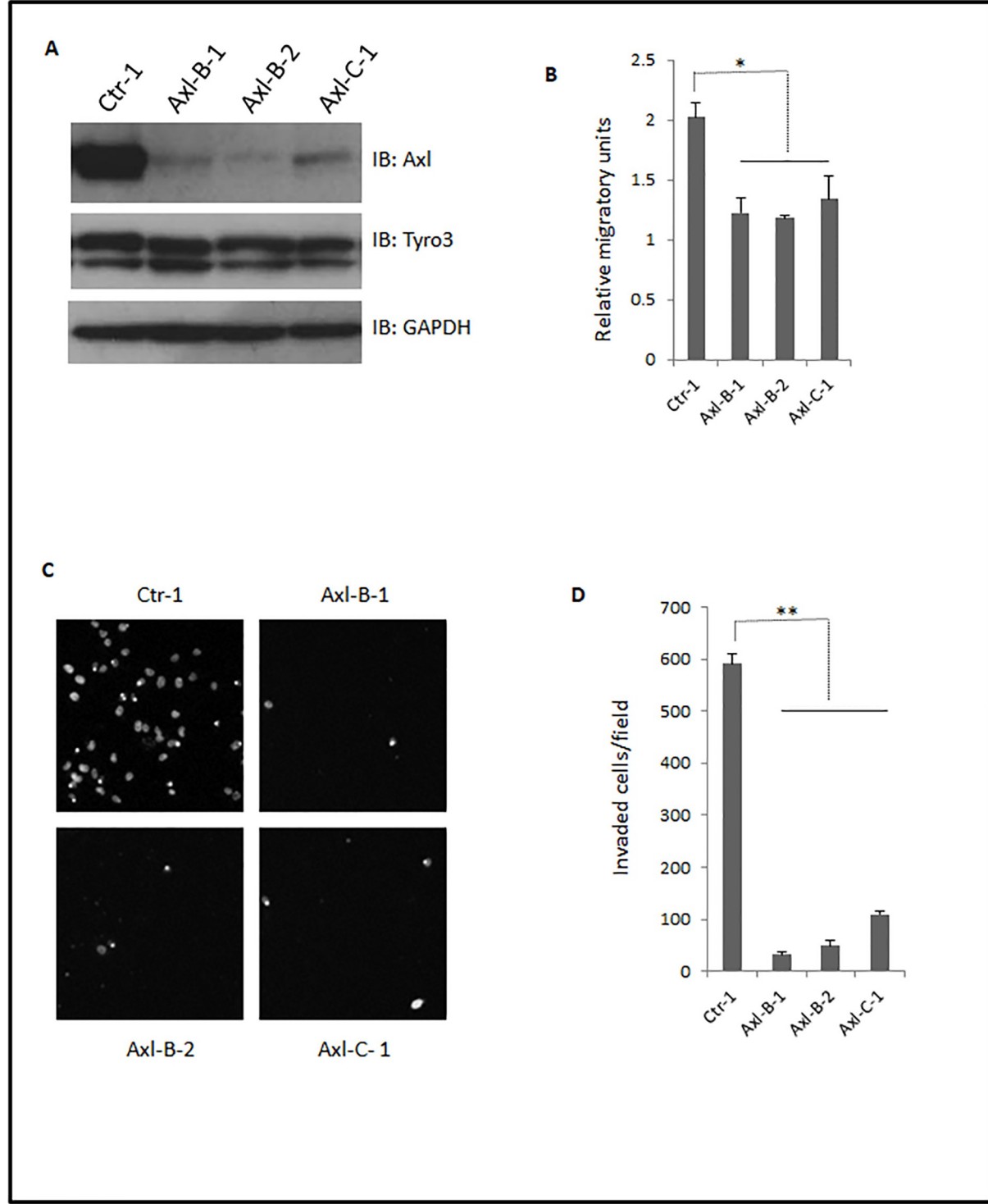

**Fig 5. Stable knockdown of Axl limits migration and invasion of IgR3 cells.** (A) Immunoblottings of stable anti-Axl shRNA expressing IgR3 monoclonal clones using indicated antibodies show selective downregulation of Axl. (B) Quantitative results of relative migratory speed of scramble and Axl shRNA clones. (C)Representative images of DAPI-stained cells invaded through MatriGel and retained on the bottom of transwell membrane. (D) Quantitative results of invaded cells. Shown are representative blots and pictographs of three experiments. Data are mean of ± SD of three independent experiments. Statistical analysis was performed using Student's t-test. * P < 0.05, ** P< 0.01, NS: not significant.

prevent the further dissemination of secondary melanomas after initial diagnoses. As we have shown that tyrosyl-directed phosphorylation of ACTN4 regulates amoeboid invasion of melanoma cells [19], we have sought the upstream regulators of this signaling pathway. As TAM family members lead to phosphorylation of ACTN4 [18], we queried whether the member most commonly associated with tumors [12], Axl, could drive melanoma progression.

In a survey of melanoma-derived cell lines we found that Tyro3 was ubiquitously present, but high levels of Axl were selective for metastatic growth phase (MGP) lines, with the exception of IgR3 (Fig 1). Thus, we investigated the role of Axl in these cells. IgR3 cells were originally isolated from a patient carrying a vertical growth phase melanoma tumor [37]. We found that stimulation of Axl with ligand could drive motility and transmigration of a barrier matrix. Finally, knockdown of Axl in IgR3 cells dramatically limited its migration and invasion *in vitro*. Unfortunately, we failed to get stable Axl shRNA WM852 cell lines when we selected it using puromycin. We noted that WM852 cells transfected with Axl shRNA grew very slowly and gradually died during selection implying that Axl is required for the viability of WM852 cells. This also suggests that Axl functions variably on the viability in each melanoma cell line.

Given that Axl can trigger the activation of Akt, ERK and p38, we tested in this study if silencing Axl by siRNA limits the basal activation of these signaling molecules. Axl siRNA did not abolish the basal activation of Akt, ERK and p38 in the absence of Gas6 stimulation but eliminated ligand-activated activation of Akt (Fig 2A). Axl has been suggested to be involved in the interaction of melanoma cells and tumor microenvironment during metastasis [12]. Thus, silencing Axl could affect these interactions resulting in an impaired invasion and metastasis. The microphthalmia transcription factor (MITF) plays a vital role in melanoma cells [5]. Previous studies revealed that most of melanoma cell lines present MITF$^{low}$/Axl$^{high}$ [5, 38]. Knockdown of Axl might affect the expression of MITF-dependent key proteins in IgR3 cells.

Our data are consistent with those of others that show that knockdown of Axl reduced the invasion of additional melanoma lines [4, 39, 40], lending further support to the contention that Axl can be target to control melanoma progression. This is in addition to the upstream targets that others have identified such as YAP [4]. In aggregate the data shown herein, demonstrating a role for Axl in promoting migration and invasion, argue for motility signaling as being key to metastatic dissemination. Thus, targeting Axl, in addition to other growth factor signaling pathways (including Braf and EGFR) [5, 41] could improve therapy aimed at preventing melanomas from metastasizing and generating lethal tumors.

## Supporting information

**S1 Fig. FBS enhances the invasion of IgR3 cells.** Quiesced IgR3 cells were invaded in quiescence media containing indicated concentration of FBS for 24h. Representative images are nuclei of invaded cells stained with DAPI.
(DOCX)

**S2 Fig. Low concentration (1uM) of R428 has no effect on the viability of melanoma cells.** Representative images of IgR3 and WM852 cells transfected with either control or Axl siRNA and treated with or without 1 μM of R428 for 24h. Shown are random images from each well.
(DOCX)

**S3 Fig. IgR3 and WM852 demonstrate variable tolerance to R428.** (A) Representative images of IgR3 and WM852 cells treated with indicated concentration of R428 for 24h. (B) Representative images of IgR3 and WM852 cells transfected with either control or Axl siRNA and treated with indicated concentration of R428 for 24h. Images were randomly taken from

each well.
(DOCX)

**S1 File.**
(PDF)

## Acknowledgments

We thank the members of the Wells lab for suggestions.

## Author Contributions

**Conceptualization:** Hanshuang Shao, Alan Wells.

**Data curation:** Hanshuang Shao, Diana Teramae.

**Funding acquisition:** Alan Wells.

**Investigation:** Hanshuang Shao, Diana Teramae.

**Methodology:** Hanshuang Shao, Diana Teramae.

**Project administration:** Alan Wells.

**Supervision:** Alan Wells.

**Writing – original draft:** Hanshuang Shao.

**Writing – review & editing:** Alan Wells.

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
