## [Decision Letter · Decision Letter 0]

1 Feb 2023

PONE-D-22-30461Axl contributes to efficient migration and invasion of melanoma cellsPLOS ONE

Dear Dr. Wells,

Thank you for submitting your manuscript to PLOS ONE. After careful consideration, we feel that it has merit but does not fully meet PLOS ONE’s publication criteria as it currently stands. Therefore, we invite you to submit a revised version of the manuscript that addresses the points raised during the review process.

Please review carefully the concerns raised by both reviewers and address each point accordingly in your revised mansucript.

We look forward to receiving your revised manuscript.

Kind regards,

Suzie Chen

Academic Editor

PLOS ONE

Journal Requirements:

"This work was supported by a Merit Award from the Veterans Administration"

"This work was supported by a Merit Award from the Veterans Administration (AW). The funders had no role in study design, data collection and analysis, decision to publish, or preparation of the manuscript."

4. We noted in your submission details that a portion of your manuscript may have been presented or published elsewhere. "A preliminary version of this manuscript has been uploaded to bioRxiv at " ext-link-type="uri" xlink:type="simple">https://www.biorxiv.org/content/10.1101/2022.05.02.490307v1" Please clarify whether this [conference proceeding or publication] was peer-reviewed and formally published. If this work was previously peer-reviewed and published, in the cover letter please provide the reason that this work does not constitute dual publication and should be included in the current manuscript.

7. We note that you have included the phrase “data not shown” in your manuscript. Unfortunately, this does not meet our data sharing requirements. PLOS does not permit references to inaccessible data. We require that authors provide all relevant data within the paper, Supporting Information files, or in an acceptable, public repository. Please add a citation to support this phrase or upload the data that corresponds with these findings to a stable repository (such as Figshare or Dryad) and provide and URLs, DOIs, or accession numbers that may be used to access these data. Or, if the data are not a core part of the research being presented in your study, we ask that you remove the phrase that refers to these data.

8. Please ensure that you refer to Figure 5 in your text as, if accepted, production will need this reference to link the reader to the figure.

Reviewers' comments:

Reviewer's Responses to Questions

**Comments to the Author**

1. Is the manuscript technically sound, and do the data support the conclusions?

Reviewer #1: Yes

Reviewer #2: Partly

2. Has the statistical analysis been performed appropriately and rigorously? 

Reviewer #1: Yes

Reviewer #2: Yes

3. Have the authors made all data underlying the findings in their manuscript fully available?

Reviewer #1: Yes

Reviewer #2: No

4. Is the manuscript presented in an intelligible fashion and written in standard English?

Reviewer #1: Yes

Reviewer #2: Yes

5. Review Comments to the Author

Reviewer #1: The manuscript by Shao and colleagues describes investigation into the effect of AXL stimulation and level on the migration and invasion of melanoma cells. The authors demonstrate clearly that stimulation of Axl signalling with Gas6 increases melanoma migration and invasion in lines with Axl expression. Further, knockdown of Axl expression either using transient or stable means reduces melanoma migration and invasion, as does treatment with a specific inhibitor. While the data arising from the stable knockdown is novel, much of the other data presented in the manuscript is confirmatory.

The manuscript overall would benefit from careful revision and editing. The experiments appear to have been conducted carefully with sufficient replication evident from the description. However, the authors should acknowledge the previous work of others in the manuscript on the topic at hand:

Sensi M, et al. (2011). "Human cutaneous melanomas lacking MITF and melanocyte differentiation antigens express a functional Axl receptor kinase." J Invest Dermatol 131:2448–57.

Use of shAXL to reduce invasion in matrigel/boyden chamber

Use of R428 to reduce migration and invasion

Tworkoski, K., et al. (2011). "Phosphoproteomic screen identifies potential therapeutic

targets in melanoma." Mol Cancer Res 9(6): 801-812.

Use of shAXL to reduce migration

Pietrobono S, Anichini G, Sala C, Manetti F, Almada LL, Pepe S, Carr RM, Paradise BD, Sarkaria JN, Davila JI, Tofani L, Battisti I, Arrigoni G, Ying L, Zhang C, Li H, Meves A, Fernandez-Zapico ME, Stecca B. ST3GAL1 is a target of the SOX2-GLI1 transcriptional complex and promotes melanoma metastasis through AXL. Nat Commun. 2020 Nov 17;11(1):5865. doi: 10.1038/s41467-020-19575-2.

Use of shAXL to reduce invasion in matrigel/boyden chamber

Zhang X, Yang L, Szeto P, Abali GK, Zhang Y, Kulkarni A, Amarasinghe K, Li J, Vergara IA, Molania R, Papenfuss AT, McLean C, Shackleton M, Harvey KF. The Hippo pathway oncoprotein YAP promotes melanoma cell invasion and spontaneous metastasis. Oncogene. 2020 Jul;39(30):5267-5281. doi: 10.1038/s41388-020-1362-9. Epub 2020 Jun 19.

PMID: 32561850

Use of shAXL to reduce invasion in matrigel/boyden chamber

In summary, the data presented in Figure 2A and B, Figure 3A-D and Figure 4A-D has been previously published by others.

Figure 2A is currently very confusing. It would be best to group the western blot images by cell line rather than antibody. Further, the figure legend states 10 nM EGF was used whereas the results text states 2 nM. Which is correct?

There is currently no text in the results section that addresses the data in Figure 5.

The reference to figures throughout the manuscript needs revision. Further, the labelling of Supplementary Figure 2 A, B - A, B, C and D is confusing and should be revised.

Reviewer #2: Eight melanoma lines were evaluated for expression of EGFR and AXL family members. IgR3 and WM852 were then stimulated with Gas6 and EGF and evaluated for pAKT, pAXL, p38, pEGFR, and pERK. Knockdown of AXL blocked Gas6 induced pAKT in both lines.as well as a line lacking AXL (WM983b) although TYRO3 was present, indicating that AXL is a mediating of Gas6 induced Akt phosphorylation. WM852 showed stronger relative migration speeds induced by Gas6 or EGF compared to IgR3. With respect to invasion, IgR3 showed higher basal migration than WM852 but there was a similar absolute increase induced by Gas6. Basal migration and invasion of the two lines was reduced either by an AXL inhibitor or siRNA.

The main claims of the paper are that AXL can mediate motility and invasion of melanoma cell lines. This has not been demonstrated previously for melanoma and thus adds to the body of knowledge for this disease.

The authors should discuss the current knowledge regarding AXL mediating Gas6 induced migration/invasion in other tumor types. There are some concerns regarding selected claims noted below. Detailed protocols are not required but more details regarding specific reagents and the initial width in the scratch assay should be indicated. The manuscript is written clearly and well organized with no dual use concerns.

Other comments:

On page 5, there is reference to “all 4 MGP” and “all 4 VGP”. Figure 1 however does not clearly indicate 4 VGP lines – please correct the inconsistency. Is WM983B a VGP, MGP or neither?

In referring to Figure 2 on page 5, it is stated” Surprisingly, we did not observe an increase in pERK in either IgR3 or WM852.” However, the blot seems to clearly show increased pERK for IgR3 in response to EGF and possibly also for GM852. Similarly, though weaker, there is EGFR phosphorylation in response to EGF although it is claimed that there is not.

The statement on page 6 in the middle : “These findings indicate that Axl signaling is cell-line specific” is confusing – one possibly interpretation is there are lines that express axl but do not respond to Gas6. Please clarify or remove.

Catalog numbers for siRNAs, antibodies, transwells, growth media, growth factors (Gas6, EGF) and all other biological reagents should be provided.

For Figure 3, please indicate the absolute basal migration speeds for all the lines. Does the smaller relative increase for IgR3 simply reflect a higher basal migration speed?

The statement at the top of page 7 “…suggest that pAkt plays important role in Gas6-mediated enhanced migration of IgR3 and WM852” does not seem justified since no inhibitor of Akt phosphorylation was tested.

Figure S2 legend refers only to panels A and B while the figure is labeled A- D.

In the discussion it is stated that ASL siRNA eliminated Gas6 induced p38 phosphorylation – that does not seem supported by Figure 2 since there is little gas6 induced p38 phosphorylation, especially in IgR3. Quantitative analysis and statistics should be used to show this point.

6. PLOS authors have the option to publish the peer review history of their article (what does this mean?). If published, this will include your full peer review and any attached files.

Reviewer #1: No

Reviewer #2: No

---

## [Author Response · Author response to Decision Letter 0]

2 Mar 2023

Journal Requirements:

Response: Please see the response below.

"This work was supported by a Merit Award from the Veterans Administration"

"This work was supported by a Merit Award from the Veterans Administration (AW). The funders had no role in study design, data collection and analysis, decision to publish, or preparation of the manuscript."

Response: It is now removed from the manuscript. Please amend to: "This work was supported by a Merit Award from the Veterans Administration (BX003368 to AW). The funders had no role in study design, data collection and analysis, decision to publish, or preparation of the manuscript."

4. We noted in your submission details that a portion of your manuscript may have been presented or published elsewhere. "A preliminary version of this manuscript has been uploaded to bioRxiv at https://www.biorxiv.org/content/10.1101/2022.05.02.490307v1" Please clarify whether this [conference proceeding or publication] was peer-reviewed and formally published. 

Response: The posted manuscript was neither peer-reviewed nor formally published.

If this work was previously peer-reviewed and published, in the cover letter please provide the reason that this work does not constitute dual publication and should be included in the current manuscript.

Response: All underlying data are presented in the manuscript and the Supporting Information files. There are no further data to post.

Response: Please state that all data are presented in the manuscript and the Supporting Information files. 

Response: Both cropped and the cognate uncropped blots are now provided in the Supporting Information files.

7. We note that you have included the phrase “data not shown” in your manuscript. Unfortunately, this does not meet our data sharing requirements. PLOS does not permit references to inaccessible data. We require that authors provide all relevant data within the paper, Supporting Information files, or in an acceptable, public repository. Please add a citation to support this phrase or upload the data that corresponds with these findings to a stable repository (such as Figshare or Dryad) and provide and URLs, DOIs, or accession numbers that may be used to access these data. Or, if the data are not a core part of the research being presented in your study, we ask that you remove the phrase that refers to these data.

Response: We now provide these data in the revised manuscript to show that the invasion of IgR3 in 1% FBS medium is significantly higher than that in 0.1% FBS medium.

8. Please ensure that you refer to Figure 5 in your text as, if accepted, production will need this reference to link the reader to the figure.

Response: We now address the data in Figure 5 in the revised text.

Response: We have updated the references as suggested by the reviewers.

5. Review Comments to the Author

Reviewer #1: The manuscript by Shao and colleagues describes investigation into the effect of AXL stimulation and level on the migration and invasion of melanoma cells. The authors demonstrate clearly that stimulation of Axl signalling with Gas6 increases melanoma migration and invasion in lines with Axl expression. Further, knockdown of Axl expression either using transient or stable means reduces melanoma migration and invasion, as does treatment with a specific inhibitor. While the data arising from the stable knockdown is novel, much of the other data presented in the manuscript is confirmatory.

Comment: The manuscript overall would benefit from careful revision and editing. The experiments appear to have been conducted carefully with sufficient replication evident from the description. However, the authors should acknowledge the previous work of others in the manuscript on the topic at hand:

Sensi M, et al. (2011). "Human cutaneous melanomas lacking MITF and melanocyte differentiation antigens express a functional Axl receptor kinase." J Invest Dermatol 131:2448–57.

Use of shAXL to reduce invasion in matrigel/boyden chamber

Use of R428 to reduce migration and invasion

Isolated melanoma cell lines

Tworkoski, K., et al. (2011). "Phosphoproteomic screen identifies potential therapeutic

targets in melanoma." Mol Cancer Res 9(6): 801-812.

Use of shAXL to reduce migration

Yale isolated melanoma cell lines

Pietrobono S, Anichini G, Sala C, Manetti F, Almada LL, Pepe S, Carr RM, Paradise BD, Sarkaria JN, Davila JI, Tofani L, Battisti I, Arrigoni G, Ying L, Zhang C, Li H, Meves A, Fernandez-Zapico ME, Stecca B. ST3GAL1 is a target of the SOX2-GLI1 transcriptional complex and promotes melanoma metastasis through AXL. Nat Commun. 2020 Nov 17;11(1):5865. doi: 10.1038/s41467-020-19575-2.

Use of shAXL to reduce invasion in matrigel/boyden chamber

A375 cell line and patient-derived melanoma cells. We actually cited this paper in ref #4

Zhang X, Yang L, Szeto P, Abali GK, Zhang Y, Kulkarni A, Amarasinghe K, Li J, Vergara IA, Molania R, Papenfuss AT, McLean C, Shackleton M, Harvey KF. The Hippo pathway oncoprotein YAP promotes melanoma cell invasion and spontaneous metastasis. Oncogene. 2020 Jul;39(30):5267-5281. doi: 10.1038/s41388-020-1362-9. Epub 2020 Jun 19.

PMID: 32561850

Use of shAXL to reduce invasion in matrigel/boyden chamber

Response: We apologize for these oversights. We now added these data and references with statement in the discussion.

Comment: In summary, the data presented in Figure 2A and B, Figure 3A-D and Figure 4A-D has been previously published by others.

Response: We include these data as part of the complete story as these are new data sets but refer to earlier works in the text. We used Gas6 to check if the interaction between Axl and EGFR is involved in the activation of Gas6-depependent of Axl in available melanoma cell lines. These data also confirmed that Gas6 functions in Axl positive melanoma cells. 

Comment: Figure 2A is currently very confusing. It would be best to group the western blot images by cell line rather than antibody. Further, the figure legend states 10 nM EGF was used whereas the results text states 2 nM. Which is correct?

Response: We now reorganized Fig. 2A by cell line according to reviewer’s suggestion. For short stimulation of cells with EGF, we usually use a concentration of 10 nM. We mistyped the concentration of EGF in text but that is now corrected.

Comment: There is currently no text in the results section that addresses the data in Figure 5.

Response: We apologize for this oversight. We now address the data in Figure 5. 

Comment: The reference to figures throughout the manuscript needs revision. Further, the labelling of Supplementary Figure 2 A, B - A, B, C and D is confusing and should be revised.

Response: Figure S2 now is reorganized, relabeled and renumbered as Figure S3 in the revised manuscript. 

Reviewer #2: Eight melanoma lines were evaluated for expression of EGFR and AXL family members. IgR3 and WM852 were then stimulated with Gas6 and EGF and evaluated for pAKT, pAXL, p38, pEGFR, and pERK. Knockdown of AXL blocked Gas6 induced pAKT in both lines.as well as a line lacking AXL (WM983b) although TYRO3 was present, indicating that AXL is a mediating of Gas6 induced Akt phosphorylation. WM852 showed stronger relative migration speeds induced by Gas6 or EGF compared to IgR3. With respect to invasion, IgR3 showed higher basal migration than WM852 but there was a similar absolute increase induced by Gas6. Basal migration and invasion of the two lines was reduced either by an AXL inhibitor or siRNA.

The main claims of the paper are that AXL can mediate motility and invasion of melanoma cell lines. This has not been demonstrated previously for melanoma and thus adds to the body of knowledge for this disease.

Comment: The authors should discuss the current knowledge regarding AXL mediating Gas6 induced migration/invasion in other tumor types. There are some concerns regarding selected claims noted below. Detailed protocols are not required but more details regarding specific reagents and the initial width in the scratch assay should be indicated. The manuscript is written clearly and well organized with no dual use concerns.

Response: We thank the reviewer for the suggestion. We now discussed this in the part of discussion of our revised manuscript. The width of the wound scratch area is about 3 mm. It is now stated in the revised manuscript.

Other comments:

Comment: On page 5, there is reference to “all 4 MGP” and “all 4 VGP”. Figure 1 however does not clearly indicate 4 VGP lines – please correct the inconsistency. Is WM983B a VGP, MGP or neither?

In referring to Figure 2 on page 5, it is stated” Surprisingly, we did not observe an increase in pERK in either IgR3 or WM852.” However, the blot seems to clearly show increased pERK for IgR3 in response to EGF and possibly also for GM852. Similarly, though weaker, there is EGFR phosphorylation in response to EGF although it is claimed that there is not.

Response: We apologize for our imprecise statements. As the labeling in Figure 1, there are three VGP cell lines and five MGP cell lines; we now corrected this in the revised manuscript. For the reference to Figure 2 on page 5, we stated that “Surprisingly, we did not observe an increase in pERK in either IgR3 or WM852” referred to no significant pERK increase when cells are treated only by Gas6. The reviewer is correct in that EGF treatment significantly increased the pERK which has been mentioned on page 6. We also stated that “the EGFR was functioning as shown by phosphorylation when exposed to 10 nM EGF for 10 min” on page 6. This meant that the EGF treatment triggers the phosphorylation of EGFR in both IgR3 and WM852 cells. It is stronger in WM852 and weaker in IgR3. These statements are now clarified.

Comment: The statement on page 6 in the middle: “These findings indicate that Axl signaling is cell-line specific” is confusing – one possibly interpretation is there are lines that express axl but do not respond to Gas6. Please clarify or remove.

Response: We thank the review for pointing out this issue. We now deleted the statement in the revised manuscript.

Comment: Catalog numbers for siRNAs, antibodies, transwells, growth media, growth factors (Gas6, EGF) and all other biological reagents should be provided.

Response: We now provide catalog numbers for all reagents.

Comment: For Figure 3, please indicate the absolute basal migration speeds for all the lines. Does the smaller relative increase for IgR3 simply reflect a higher basal migration speed?

Response: The reviewer is correct in that IgR3 presents much high basal migration. We now updated Figure 3C with relative migratory units instead of normalized data to more clearly show this.

Comment: The statement at the top of page 7 “…suggest that pAkt plays important role in Gas6-mediated enhanced migration of IgR3 and WM852” does not seem justified since no inhibitor of Akt phosphorylation was tested.

Response: We thank the reviewer for pointing out this misstatement. We now corrected it to “suggest that Axl plays an important role in Gas6-mediated enhanced migration of IgR3 and WM852.

Comment: Figure S2 legend refers only to panels A and B while the figure is labeled A- D.

Response: We thank the reviewer for noting this omission. We now reorganized this figure and corrected the figure legend as shown in figure S3 in the revised manuscript.

Comment: In the discussion it is stated that ASL siRNA eliminated Gas6 induced p38 phosphorylation – that does not seem supported by Figure 2 since there is little gas6 induced p38 phosphorylation, especially in IgR3. Quantitative analysis and statistics should be used to show this point.

Response: We thank the reviewer for noting the lack of significance of this and have corrected this in the revised manuscript.

---

## [Decision Letter · Decision Letter 1]

15 Mar 2023

Axl contributes to efficient migration and invasion of melanoma cells

PONE-D-22-30461R1

Dear Dr. Wells,

We’re pleased to inform you that your manuscript has been judged scientifically suitable for publication and will be formally accepted for publication once it meets all outstanding technical requirements.

Kind regards,

Suzie Chen

Academic Editor

PLOS ONE

Additional Editor Comments (optional):

Reviewers' comments:

Reviewer's Responses to Questions

**Comments to the Author**

1. If the authors have adequately addressed your comments raised in a previous round of review and you feel that this manuscript is now acceptable for publication, you may indicate that here to bypass the “Comments to the Author” section, enter your conflict of interest statement in the “Confidential to Editor” section, and submit your "Accept" recommendation.

Reviewer #1: All comments have been addressed

Reviewer #2: All comments have been addressed

2. Is the manuscript technically sound, and do the data support the conclusions?

Reviewer #1: (No Response)

Reviewer #2: (No Response)

3. Has the statistical analysis been performed appropriately and rigorously? 

Reviewer #1: (No Response)

Reviewer #2: (No Response)

4. Have the authors made all data underlying the findings in their manuscript fully available?

Reviewer #1: (No Response)

Reviewer #2: (No Response)

5. Is the manuscript presented in an intelligible fashion and written in standard English?

Reviewer #1: (No Response)

Reviewer #2: (No Response)

6. Review Comments to the Author

Reviewer #1: (No Response)

Reviewer #2: (No Response)

7. PLOS authors have the option to publish the peer review history of their article (what does this mean?). If published, this will include your full peer review and any attached files.

Reviewer #1: No

Reviewer #2: No

---

## [Editor Report · Acceptance letter]

20 Mar 2023

PONE-D-22-30461R1 

Axl contributes to efficient migration and invasion of melanoma cells 

Dear Dr. Wells:

I'm pleased to inform you that your manuscript has been deemed suitable for publication in PLOS ONE. Congratulations! Your manuscript is now with our production department. 

Kind regards, 

on behalf of

Dr. Suzie Chen 

Academic Editor

PLOS ONE